# Efficacy and Safety of Botulinum Toxin B in Focal Hyperhidrosis: A Narrative Review

**DOI:** 10.3390/toxins15020147

**Published:** 2023-02-11

**Authors:** Anna Campanati, Federico Diotallevi, Giulia Radi, Emanuela Martina, Barbara Marconi, Ivan Bobyr, Annamaria Offidani

**Affiliations:** Dermatological Clinic, Department of Clinical and Molecular Sciences, Polytechnic Marche University, 60121 Ancona, Italy

**Keywords:** botulinum toxin type B, focal hyperhidrosis, palmar hyperhidrosis, axillary hyperhidrosis, craniofacial hyperhidrosis, plantar hyperhidrosis, compensatory hyperhidrosis, residual hyperhidrosis

## Abstract

Botulinum toxin type B (BoNT-B), known as Myobloc^®^ in the United States and as Neurobloc^®^ in Europe, is a new therapeutically available serotype among the botulinum toxin family. During the last years several data have been reported in literature investigating its efficacy and safety, as well as defining the dosing and application regiments of BoNT-B in the treatment of hyperhidrosis. Moreover, recent studies have been examining its safety profile, which may be different from those known about BoNT-A. The aim of this review is to provide information about what is currently known about BoNT-B in regards to the treatment of focal hyperhidrosis.

## 1. Introduction

The burden of focal hyperhidrosis is directly related to its social, psychological and occupational repercussions, which make it a distressing and disabling condition for affected patients. 

Although the use of botulinum toxin type A (BoNT-A) has radically changed the management of patients suffering from axillary and palmar hyperhidrosis, and it has gained a positioning in the guidelines of the International Hyperhidrosis Society, several limiting factors related to duration of benefit, and hand muscle weakening related to motor nervous system involvement mainly following BoNT-A diffusion through thenar and hypothenar hand eminence have driven several authors to investigate the efficacy and safety profile of botulinum toxin type B (BoNT-B) for the treatment of focal hyperhidrosis [1].

BoNT-B is produced as relatively inactive, single polypeptide chains with a molecular mass of about 150 kDa consisting of a heavy (H) chain and a light (L) chain of roughly 100 and 50 kDa, respectively, linked by a disulfide bond [2]. The botulinum toxin neurotoxin complex is also associated with various other nontoxic proteins, which may also have hemagglutinating properties.

The efficacy of BoNT-B in focal hyperhidrosis is related to its interfering action on neural transmission by blocking the release of acetylcholine. BoNT-B inhibits acetylcholine release at the neuromuscular junction via a three-stage process: (1) H-chain-mediated neurospecific binding of the toxin, (2) internalization of the toxin by receptor-mediated endocytosis, and (3) ATP and pH dependent translocation of the L chain to the neuronal cytosol where it acts as a zinc-dependent endoprotease cleaving polypeptides essential for neurotransmitter release.

BoNT-B binds to and cleaves the synaptic Vesicle Associated Membrane Protein (VAMP, also known as synaptobrevin) which is a component of the protein complex responsible for docking and fusion of the synaptic vesicle to the presynaptic membrane, a necessary step for neurotransmitter release [3].

BoNT-B is available in a ready-to-use formula that does not require reconstitution. However, many physicians choose to dilute it to reduce the pain on injection. When diluting a vial of BTX-B, it is important to consider the bottles are overfilled and actually contain slightly more toxin than the label states, to account for excess lost in the needle tip. BoNT-B is available in three vial configurations of 2500 U, 5000 U, and 10,000 U, with a composition of 5000 U BoNT-B/mL. BoNT-B vials are stable for 30 months when refrigerated and for 9 months at room temperature [4].

The aim of this narrative review is to provide an update on the efficacy and safety profile of BTX-B in the treatment of focal hyperhidrosis.

## 2. Results

A proof of efficacy of BoNT-B on sweat production was described in 2003 by Birklein F et al. [5]. The authors analyzed the suppression of sweat gland activity by BoNT-B by injecting it (between 2 and 1000 mouse units subcutaneously) in the lateral side of both lower legs in 15 healthy volunteers. Sweat tests were carried out before BoNT-B injections, and at 3, 12, and 24 weeks, by iodine-starch staining and by capacitance hygrometry after carbachol iontophoresis, according to the quantitative sudomotor axon reflex test (QSART). Iodine starch staining indicated that a threshold dose of 8 MU BoNT-B leads to anhidrotic skin spots (>4 cm^2^) after 3 weeks. The duration of anhidrosis was prolonged for 12 weeks when 15 MU BoNT-B was injected, and for 24 weeks when 125 MU BoNT-B was injected. The size of the anhidrotic area decreased with time, indicating partial recovery at the edges. After 3 weeks, the QSART score had significantly decreased to 18% of the baseline and had decreased to zero in most subjects with doses of 62.5 MU or more. After 12 weeks, the QSART had returned to 91% of the baseline in all but one subject and, after 24 weeks, recovery of sudomotor function was complete. Analysis by iodine-starch staining and QSART indicated that BoNT-B suppresses sudomotor function effectively in a concentration-dependent manner.

### 2.1. BoNT-B in Axillary Hyperhidrosis

Results reported in Table 1. In 2002, Dressler et al. [6] firstly reported their experience in the use of BoNT-B for the treatment of bilateral axillar hyperhidrosis, through a direct comparison of the antihyperhidrotic effect of BoNT-B (NeuroBloc^®^/MyoBloc^®^) with that of BoNT-A (Botox^®^). Nine patients (High Dose group-HD) received BoNT-A 100MU unilaterally and BoNT-B 4000 MU contralaterally. Ten patients (Low Dose group-LD) received BoNT-A 100 MU and BoNT-B 2000 MU. All patients were blinded as to which preparation was used in which side. All patients except one reported excellent hyperhidrosis improvement in both axillae. None of the patients had residual symptoms on clinical examination. The duration of hyperhidrosis improvement until first recurrence in the HD group was similar between the BoNT-B and BoNT-A sides. In the LD group, duration was prolonged in the BoNT-A side. There was no difference in the duration of hyperhidrosis improvement between the axillae treated with BoNT-B 4000 MU and BoNT-B 2000 MU. Five out of nine patients in the HD group and seven out of ten patients in the LD group reported more application discomfort in the BoNT-B treated axillae. In six out of nine patients in the HD group, and in six out of ten patients in the LD group, the onset of hyperhidrosis improvement appeared earlier in the BoNT-B treated axillae. One patient in the HD group reported dryness of the mouth and eyes and accommodation difficulties. These authors viewed BoNT-B as a safe and efficient treatment for axillary hyperhidrosis. Doses of BoNT-B 2000 MU per axilla seemed sufficient, indicating a conversion factor between BoNT-A and BoNT-B in the order of 1:20, as the autonomic nervous system seems to be relatively more sensitive to BoNT-B than to BoNT-A compared with the motor system.

In 2004, Hecht MJ et al. [7] reported their experience in the use of BoNT-B for axillary hyperhidrosis in four patients. Axillary sweating of the patients was measured by gravimetry and the hyperhidrotic area was visualized by iodine-starch staining and 250 MU BoNT-B diluted in 2.5 mL of saline was injected subcutaneously into each axilla at 10–15 injection sites. The patients were re-evaluated 3 and 12 weeks later. The therapeutic effects were assessed both gravimetrically and by iodine-starch staining, and the patients estimated the therapeutic effects by subjective rating. Three weeks after the BoNT-B-injection, three of four patients had axillary anhidrosis, and in the fourth patient, axillary sweating was dramatically reduced. All three patients with total blockage of axillary sweating assessed the result as “excellent”, and the patient with reduced sweating assessed the result as “fair”. The duration of sweat suppression ranged from 1 to 3 months. After 3 months axillary sweating recurred in all patients. The dose used in reported study was very low (250 MU), without clinically apparent systemic side effects, but it also an insufficient dose to block sweating completely in patients with very excessive sweating. 

In the same year, Nelson L et al. [8] investigated the efficacy of BoNT-B (Neurobloc^®^) for axillary hyperhidrosis. Thirteen patients (22 axillae) were recruited to the study. The hyperhidrotic area was defined using the iodine-starch test then measured and photographed. A total of 5000 MU of BoNT-B was administered subdermally. Patients were reviewed at 4, 8, and 12 weeks to assess the outcome objectively (using hyperhidrotic area measurements and photographs) and subjectively (by assessing sweat production and patient satisfaction). A significant reduction in the hyperhidrotic area at follow-up compared to the baseline was demonstrated. The mean percentage reduction in the hyperhidrotic area was 84, 87, and 81% at 4, 8, and 12 weeks, respectively (*p* = 0.001, paired t test). Patient satisfaction was 100% throughout. Subjective mean percentage reduction in sweat production was 98, 96, and 90% at 4, 8, and 12 weeks, respectively. Side effects were negligible.

In 2005, Baumann L et al. [9] reported preliminary results on the use of BoNT-B in axillary hyperhidrosis, through a double-blinded, randomized pilot study conducted in an outpatient office setting at a private academic medical center beginning in November 2001. Twenty subjects with primary axillary hyperhidrosis were enrolled and injected subcutaneously with either BoNT-B (Myobloc^®^) (2500 U, or 0.5 mL, per axilla) or 0.5 mL vehicle (100 mM NaCl, 10 mM succinate, and 0.5 mg/mL human albumin) into bilateral axillae. Participants who received a placebo were rolled over and received BoNT-B at subsequent visits. All participants were followed until sweating returned to baseline levels. According to the participant assessment of axillary hyperhidrosis improvement and quality of life scores and the physician assessment scores, a significant difference was observed in treatment response at day 30 in the participants receiving Myobloc (BoNT-B (Myobloc^®^)) injections. The duration of action ranged from 2.2 to 8.1 months (mean 5.0 months). The adverse event profile included bruising, flu-like symptoms, and dry eyes. 

In 2011, Frasson E et al. [10] compared the antihyperhidrotic effect of intra-axillary injections of BoNT-B (Botulinum toxin-Myobloc^®^) with BoNT-A (Onabotulinum toxin-Botox^®^) to address the question concerning the effective dose. They performed a bilateral paired, single-blinded, randomized study including 10 patients with idiopathic focal axillary hyperhidrosis since childhood that was unresponsive to other nonsurgical treatments, who received (BoNT-A) unilaterally and BoNT-B contralaterally. 

Each patient was injected in one axilla with 50 U of (BoNT-A) diluted with 1 mL of 0.9% sterile physiologic saline without preservatives and in the contralateral axilla with 2500 U of BoNT-B diluted with 0.5 mL of 0.9% sterile physiologic saline without preservatives. The identified hyperhidrotic area was pen-marked and subdivided into 2 × 2-cm square (4 cm^2^).

The toxin was injected in amounts of 0.025 mL (BoNT-B) and 0.050 mL (BoNT-A) intradermally. All patients received the same amount of toxins divided among the same injection points. 

All patients reported a reduction in axillary sweat production. Mean pretreatment sweat production rates and areas were similar bilaterally. After BoNT injections, patients responded to treatment until month six. At 1 and 2 weeks and 1, 3, and 6 months after treatment, sweat weight and area decreased significantly more in the BoNT-B side than in the BoNT-A side. Moreover, patients’ treatment satisfaction scores were significantly higher for the BoNT-B than for the BoNT-A treatment until month three, and treatment began acting earlier in the BoNT-B side compared to the A/Ona side. All patients tolerated intradermally injected BoNTs, although some experienced mild pain, especially during BoNT-B injections. No hematomas developed at the injection site, nor did any participant report systemic adverse effects. The authors provided objective evidence that BoNT-B is safe and effective for treating bilateral axillary hyperhidrosis and, when administered at the same dose ratio of 1:50 used for the motor system, it blocks sweating better than BoNT-A. 

### 2.2. BoNT-B in Palmar Hyperhidrosis

Results reported in Table 2. Baumann L. et al in 2005 [11] reported their experience in the use of BoNT-B (Myobloc^®^) for palmar hyperhidrosis, through a double-blind, randomized, placebo-controlled study, including twenty patients (10 men and 10 women) diagnosed with palmar hyperhidrosis, who were injected with either Myobloc^®^ (5000 U per palm) or a 1.0 mL vehicle (100 mM NaCl, 10 mM succinate, and 0.5 mg/mL human albumin) into bilateral palms (15 patients received Myobloc^®^ and 5 received the placebo). The participants were followed until sweating returned to baseline levels. The main outcome measures were safety, efficacy versus placebo, and the duration of effect. A significant difference was found in treatment response at day 30, as determined by participant assessments, between 15 participants injected with Myobloc^®^ and 3 participants injected with the placebo. The duration of action ranged from 2.3 to 4.9 months, with a mean duration of 3.8 months. The single most-reported adverse event was dry mouth or throat, which was reported by 18 of 20 participants. The adverse event profile also included indigestion or heartburn (60%), excessively dry hands (60%), muscle weakness (60%), and decreased grip strength (50%). The authors stated that Myobloc^®^ proved to be effective for the treatment of palmar hyperhidrosis, and it had a rapid onset, with most participants responding within 1 week. The duration of action ranged from 2.3 to 4.9 months, with a mean of 3.8 months. The adverse event profile included dry mouth, indigestion or heartburn, excessively dry hands, muscle weakness, and decreased grip strength, as already described in the literature. 

In 2013, Rosell K et al. [12] investigated the effect of a mixed treatment (BoNT-A-Xeomin^®^ plus BoNT-B toxin-Neurobloc^®^) of palmar hyperhidrosis on patients’ quality of life. Twenty-six patients with palmar hyperhidrosis were injected with 213 ± 19 U Xeomin^®^ and 264 ± 60 U Neurobloc^®^ over the thenar eminences to avoid muscle weakness. At follow-up 3 weeks post-treatment, 95% of patients were satisfied, evaporation decreased >50% and the Dermatology Life Quality Index (DLQI) score significantly improved from 10.3 to 1.2. Only one patient experienced muscle weakness. In conclusion, Xeomin^®^ in combination with Neurobloc^®^ has an excellent effect on palmar hyperhidrosis, and Neurobloc^®^ may be an option for use in the treatment of palmar hyperhidrosis in order to minimize muscular side effects.

In 2014, Basciani M. et al. [13] reported their experience in the use of BoNT-B in treating primary palmar hyperhidrosis. Participants were injected with 5000 IU of BoNT-B in each palm. A visual analogue test (VAS) was used to evaluate the intensity of the decrease in sweat production; Minor’s iodine-starch test and a measurement of paper towels’ weight were used to ascertain palmar sweating at the baseline, 4, 12, and 24 weeks after BoNT-B injections by a blind examiner. Thirty-two subjects (12 males, 20 females, mean age 31 ± 11) were enrolled. A significant reduction of palmar sweating was detected after BoNT-B injection: 2.9 ± 1.4, 0.3 ± 0.4, 0.9 ± 0.8, and 2.1 ± 1.5 g (*p* < 0.001) of paper towels’ weight for the right palm at the baseline, 4, 12, and 24 weeks; and 2.8 ± 1.7, 0.5 ± 0.6, 0.8 ± 0.7, and 1.8 ± 1.25 g (*p* < 0.001) at same time, respectively, for the left palm. A significant reduction of mean VAS values were also detected after BoNT-B injections: 8.6 ± 1.1, 0.6 ± 0.8, 3.5 ± 2.5, and 7.1 ± 2.4 (*p* < 0.0001) at the baseline, 4, 12 and 24 weeks, respectively. Mild side effects consisting of local pain and hand weakness were observed in four (12.5%) subjects. The authors concluded that the use of 5000 IU BoNT-B injection in each palm was safe and significantly improved the severity of palmar hyperhidrosis.

### 2.3. BoNT-B in Plantar Hyperhidrosis

No case report, case series, original prospective or retrospective studies have been reported concerning the use of BoNT-B for plantar hyperhidrosis, thus BoNT-A still represents the only serotype used for this site, with good results over the short-term, but unsatisfactory long-term results [14]. 

### 2.4. BTX-B in Cranio-Facial Hyperhidrosis

Results reported in Table 3. In 2014, Karlqvist M. et al. [15] reported a prospective open study on the use of BoNT-B for cranio-facial hyperhidrosis. Thirty-eight consenting patients with craniofacial hyperhidrosis were consecutively enrolled to primarily evaluate the clinical effect of BoNT-B in craniofacial hyperhidrosis 2–4 weeks post-treatment and secondarily to investigate the duration of the therapeutic effects and safety of the treatment. The primary endpoint was the difference in total DLQI score before and 2–4 weeks after treatment. The secondary endpoints were as follows: the difference in sweat production before and 2–4 weeks after treatment monitored with transepidermal water loss (g/m^2^/h) and gravimetry (mg/min), the time to reinjection as a measure of duration of the treatment effect and the frequency of adverse events which were captured throughout the study. BoNT-B (Neurobloc^®^ 250 U/mL, Eisai Co., Ltd. (Tokyo, Japan)) was used and 5 U per injection were administered intradermally every 15 mm in a checked pattern covering the entire hyperhidrotic area, taking special care to inject the area of the forehead no closer to the eyebrows than 40 mm. Injections of hyperhidrotic areas were not limited to the face but the forehead, neck, and parts of the scalp (tonsurans) were also treated. DLQI scores were significantly improved at follow-up 2–4 weeks post-treatment and sweating was significantly reduced. Similarly, sweating before treatment monitored with transepidermal water loss and gravimetry significantly decreased after treatment. Regarding the Global Assessment of Therapy, 87% of the patients were satisfied with the treatment result. In a 2-year follow-up, 74% returned for further treatment after a median time of 5 months. Side effects were mild and the most-reported was stiffness of the forehead and the eyebrows. The authors concluded that BoNT-B seemed to be both a safe and effective treatment in craniofacial hyperhidrosis improving quality of life and reducing extreme sweating.

In 2019, Cabreus P et al. [16] described a randomized controlled trial regarding BoNT-B treatment in craniofacial hyperhidrosis. Eight postmenopausal patients were randomized to receive BoNT-B or a placebo. Measurements were performed before treatment and 3 ± 1 weeks after. The DLQI score was improved for all patients after BoNT-B treatment (n = 3) with a median decrease of nine points (90% median improvement). The placebo group (n = 5) had a median increase of two points (−18% median decline). When the same group (n = 5) received BoNT-B (open), the DLQI score decreased, with a median of seven points compared with the baseline (91% median improvement). Treatment-related adverse events were temporary and did not prevent the improvement of quality of life. The results indicated that BoNT-B seemed to be a safe and effective treatment in postmenopausal craniofacial hyperhidrosis, although further research is encouraged.

Cantarella G et al. [17] in 2010 reported their experience in the use of BoNT-B in Frey’s syndrome.

Frey’s syndrome is a frequent sequela of parotidectomy, causing facial sweating and flushing because of gustatory stimuli. Although botulinum toxin type A has become a first-line therapy for Frey’s syndrome, some patients become resistant, thus the authors investigated whether another serotype, BoNT-B, might be an effective alternative. Seven patients aged 30 to 68 years, with severe Frey’s syndrome, underwent the Minor test and had 80 U of BoNT-B B per cm^2^ (mean total dose, 2354 U) injected intracutaneously in the mapped area of gustatory sweating. All patients were followed up for 12 months. One month after treatment, six of the seven patients reported that gustatory sweating and flushing had resolved, and, in the remaining patient, these symptoms had decreased. The Minor test confirmed a significant improvement. The subjective benefits remained stable for six months in four patients and for nine months in the remaining three patients; 12 months after treatment, all patients still reported some improvement. The authors concluded that BoNT-B afforded symptomatic relief in a small sample of patients with Frey’s syndrome and might be considered a potential alternative to botulinum toxin type A.

### 2.5. BoNT-B in Compensatory and Residual Hyperhidrosis 

#### 2.5.1. Compensatory Hyperhidrosis

Results reported in Table 4. Compensatory hyperhidrosis (CH) is the most common adverse complication of endoscopic transcutaneous sympathectomy (ETS). The prevalence of CH is reported in 30–100% of patients who have gone through ETS [18]. Most often, 1–6 months postoperatively, excessive sweating appears after little or no effort below the denervation zone, especially on the abdomen [19].

A major problem in the treatment of CH is that patients typically have large areas with excessive sweating which limit the use of BoNT-A. However, BoNT-B has a greater affinity for sudomotor fibers and alleviates excessive sweating from large areas in relatively small doses [5,6,20,21]. Karlsson-Groth A et al. [22] described their experience in the use of BoNT-B in CH. Among nine consecutively included patients, aged 36–69 years, six male and three female, with CH after sympathectomy, followed at the Sweat Clinic of Sophiahemmet, Stockholm, between the 1st of September to the 31st of October 2009, seven were treated with BoNT-B (Myobloc^®^/NeuroBloc^®^) at the trunk, with dosage raging from 1000 to 4000 MU. NeuroBloc^®^/Myobloc^®^ 250 U/mL, was given every 15 mm with 7.5 U at each spot. The patients responded to a DLQI questionnaire before injections with BoNT-B and 3 weeks after treatment. At the follow-up visit, the participants also ranked the effect of the treatment on a five-grade scale. Three patients had residual sweating after BoNT-B treatment, and received additional anticholinergics at the follow-up visit. Those subjects eventually had a third evaluation with the DLQI. The DLQI score was, on average, 16.4 before treatment and decreased to 4.8 after BoNT-B injections. Eight out of nine patients were satisfied with the treatment. The average DLQI score decreased to 2.2 when the patients with residual sweating (n = 3) received additional anticholinergics. Adverse events from BoNT-B were mild and temporary, but dry mouth was substantial in one patient using anticholinergics. For the authors, a combination of BTX A/B and anticholinergics alleviated the hyperhidrosis with minor side-effects. We consider this treatment safe, effective, and well tolerated.

**Table 4 toxins-15-00147-t004:** BoNT-B in compensatory hyperhidrosis data from the literature.

First Author Year [Ref.]	Type of Study	Patients [n]	BoNT-B Doses	Follow Up	Results
Karlsson-Groth A et al., 2015 [22]	Prospective open study	9	BoNT-B 1000 to 4000 MU	3 weeks	Eight out of nine patients were satisfied with the treatment.

#### 2.5.2. Residual Hyperhidrosis

Results reported in Table 5. Focal hyperhidrosis can also frequently occur after limb amputation, where the remaining residual limb excessively perspires, leading to an increased risk of dermatological disorders and functional limitations, such as the inability to wear a prosthesis comfortably or safely. Although many treatments have been proposed to treat residual hyperhidrosis within the dermatology community, they are not widely known by healthcare providers typically involved in caring for individuals with acquired limb loss [23]. 

In 2006, Pasquina PF et al. [24] investigated the use of (BoNT-B [Myobloc^®^]) compared with a placebo in treating hyperhidrosis in the residual limbs of individuals with amputation in a randomized, double-blind, placebo-controlled pilot study. Nine male patients with eleven major amputations of the lower limbs and who complained of excessive sweating in their residual limbs were enrolled in the study between 24 September 2008 to October 28, 2011. Participants’ lower limbs were randomly assigned to receive injections of either BoNT-B (n = 7) or a placebo (n = 4). The primary efficacy variable was a minimum of 50% reduction in sweat production 4 weeks after the injection as measured via gravimetric sweat analysis after 10 min of physical exertion. Secondary analyses were performed on prosthetic function and pain. All volunteers (100%; seven) in the BoNT-B group achieved a minimum of 50% reduction in sweat production as compared with only 50% (two) in the placebo group. The percent reduction was significantly greater for the BoNT-B group than for the placebo group (−72.7% ± 15.7% vs −32.7% ± 39.2%; *p* < 0.005). Although both groups subjectively self-reported significant sweat reduction and improved prosthetic function (*p* < 0.05 for both), objective gravimetric sweat analyses significantly decreased only for the BoNT-B group (2.3 ± 2.3 g vs 0.7 ± 1.1 g; *p* < 0.005). Neither group reported a change in phantom limb pain or residual limb pain (*p* > 0.05 for both). BoNT-B successfully reduces sweat production in individuals with residual limb hyperhidrosis, but does not affect pain. 

**Table 5 toxins-15-00147-t005:** BoNT-B in residual hyperhidrosis data from the literature.

First Author Year [Ref.]	Type of Study	Patients [n]	BoNT-B Doses	Follow Up	Results
Pasquina PF et al., 2005 [24]	Randomized, double-blind, placebo-controlled pilot study	9	BoNT-B 1000 to 4000 MU	4 weeks	The percent reduction was significantly greater for the BoNT-B group than for the placebo group.
Kern U et al., 2011 [25]	Pilot prospective study	9	BoNT-B 1750 MU	4–12 weeks	Significant improvements to sweating of the residual limb 4 weeks after BoNT-B treatment. Steadiness of gait, quality-of-life, and work performance increased accordingly, and skin problems decreased clearly but not significantly.

These results are reinforced by those of Kern U et al. in 2011 [25]. They performed a pilot study to prove the alleviating effect of BoNT-B in sweating of the residual limb, as BoNT-B even in low doses, is supposed to possess a more specific action in sympathetic nerves than botulinum toxin type A does at a wider radius of diffusion. Nine lower limb amputees received 1750 U BoNT-B injected at the site of maximum sweating. Before injections and 4 weeks and 3 months after, patients rated their impairments regarding sweating of the residual limb, steadiness of gait, use of the prosthetic device, quality-of-life, work performance, quality of sleep, and skin problems using a numeric rating scale (NRS; 0–10). Sweating of the residual limb before BoNT-B application was rated at a median of seven (interquartile range, 6–10) on the NRS, with significant improvements after 4 weeks (NRS, 3 (2–4); *p* = 0.027) and 3 months (NRS, 3 (1–4); *p* = 0.020). Impaired quality of artificial limb use likewise improved from a baseline NRS of nine (5–9) to two (1–4) after 4 weeks and three (1–4) (*p* = 0.027) after 3 months, consistent with limited duration of use (*p* = 0.023). Steadiness of gait, quality-of-life, and work performance increased accordingly, and skin problems decreased clearly but not significantly. Unexpectedly, stump pain was also reduced (baseline: NRS, five (4–8); 4 weeks: NRS, four (3–5), *p* = 0.109; 3 months: NRS, three (2–4), *p* = 0.008). For these authors, a low-dose of BoNT-B significantly reduced sweating of the residual limb, thereby improving the use of the artificial limb, steadiness of gait, and quality of life.

## 3. Discussion and Conclusions

Since the discovery of the therapeutic potential of botulinum toxin for the treatment of strabismus in 1980, its therapeutic applications has grown dramatically over the years. Among the BoNTs family, BoNT-A represents the most well-known and well-studied, and it now provides therapy for patients in neurology, ophthalmology, pediatrics, surgery, and dermatology [26].

A new serotype of botulinum toxin has arrived in the United States. Botulinum toxin type B (BoNT-B), known as Myobloc^®^ in the United States and as NeuroBloc^®^ in Europe. 

Botulinum Toxin Type B inhibits acetylcholine release at the neuromuscular junction via a three-stage process: (1) Heavy-chain-mediated neurospecific binding of the toxin, (2) internalization of the toxin by receptor-mediated endocytosis, and (3) ATP and pH dependent translocation of the light chain to the neuronal cytosol where it acts as a zinc-dependent endoprotease cleaving polypeptides essential for neurotransmitter release.

Botulinum Toxin Type B binds to and cleaves the synaptic Vesicle Associated Membrane Protein (VAMP, also known as synaptobrevin) which is a component of the protein complex responsible for docking and fusion of the synaptic vesicle to the presynaptic membrane, a necessary step for neurotransmitter release [27].

BoNT-A and BoNT-B have been used for a myriad of dermatologic issues in an “off-label” way [28]. It has followed in the therapeutic footsteps of BTX-B in treating focal hyperhidrosis. Over the past years, researchers have been increased knowledge about the efficacy and safety of BoNT-B (which are quite different to that seen with BoNT-A) in focal hyperhidrosis, as well as defined dosing and application regimens of BoNT-B for treating hyperhidrosis.

Axillary hyperidrosis represents the most-studied application for BoNT-B in focal hyperhidrosis, with five reported studies including 57 patients treated with BoNT-B with dosage ranging from 1000 to 5000 MU. Taken together, data on efficacy and its safety profile indicate that BoNT-B has a rapid onset of action, with most participants responding with a cessation of sweating in 5–7 days. The duration of action ranged from 2.2 to 8.1 months, with a mean of 5.0 months. The adverse event profile is generally acceptable and includes bruising and dry eye (in about 25% of subjects), dry mouth (in 35%), and flu-like symptoms (in 30%). 

Less studies concerning the use of BoNT-B in palmar hyperhidrosis have been reported, although the total number of patients with palmar hyperhidrosis receiving a treatment with BoNT-B was greater for palmar than axillary hyperhidrosis (78 vs. 57). 

The anhidrotic effect of BoNT-B largely depends on the dose used, with significant reduction of palmar sweating after injection of BTX-B at a dose of 5000 U per palm ranging from 2.3 to 4.9 months, with a mean duration of 3.8 months. The most commonly reported adverse event was dry mouth/throat, which is reported by almost 90% of the patients. The adverse effect profile also includes indigestion/heartburn (in 60%), excessively dry hands (in 60%), muscle weakness (in 60%), and decreased grip strength (in 50%). BoNT-B also had a rapid onset, with most participants responding within 1 week.

More studies evaluating the use of BTX-B to manage both axillary and palmar hyperhidrosis are clearly required, mostly because there is no consensus yet on optimal dosing regiments for using BTX-B as well as conversion factors for converting BoNT-A to BoNT-B [4].

Few data have been reported on craniofacial hyperhidrosis treatment with BoNT-B, including 53 patients (seven among them with Frey’s syndrome). The overall results indicate BoNT-B to be a safe and effective treatment in craniofacial hyperhidrosis, with no different emerging side effects, compared to axillary hyperhidrosis.

With regard to compensatory hyperhidrosis, the greatest challenge consists in its treatment, as it is usually diffuse on larger areas than focal hyperhidrosis, so that BoNT-A at standardized dosage is of limited efficacy. For this reason, BoNT-B, which has a greater affinity for sudomotor fibers, could alleviate excessive sweating on large areas with relatively small doses, and preliminary data seem to confirm this specific valence of BoNT-B.

Finally, preliminary data have been reported about the use of BoNT-B for treatment of residual hyperhidrosis in amputated patients. The impact of focal hyperhidrosis is tremendous in amputees, and we are in dire need for systematic, larger studies to confirm the preliminary results of the efficacy of BoNT-B. 

Two major concerns are related to BoNT-B use in focal hyperhidrosis: the increasing incidence in the severity of related side effects and the increased pain at injection site, compared to BoNT-A. BoNT-B has been shown to produce more systemic side effects than BoNT-A and this is indicative of a difference between BoNT-B and BoNT-A. It has been hypothesized that BoNT-B may have a higher affinity for autonomic nerve endings than BoNT-A, although the mechanism of action underlying this affinity remains only speculative. However, BoNT-B should be used with caution in patients with pre-existing autonomic dysfunction and in those with conditions for which anticholinergic therapy is contraindicated [29].

On the other hand, the postulated preferential action of BoNT-B on the autonomic nervous system, could give BoNT-B a greater specificity of use in the management of palmar hyperhidrosis, with a hypothetical potential lower risk of generating hyposthenia, compared with BoNT-A.

To verify if these assumptions are real and BoNT-B merits increasing use in palmar hyperhidrosis to guarantee a long-lasting benefit without hand muscle weakening, the increase in experience of use is mandatory. 

Another specific side effect that becomes more and more apparent with increasing clinical use of BoNT-B is injection site pain. The incidence of injection site pain appears to be higher for BoNT-B than for BoNT-A. The reason that BoNT-B is less comfortable than BoNT-A remains unknown, although several hypotheses have been considered; the greater volume that is injected, the concentration of active or inactive proteins, or even the lower pH of the solution which is required to stabilize the BoNT-B solution. The latter hypothesis seems to be the most accredited. Diluting the original BoNT-B preparation with saline to make 250 MU in 2.5 mL also dilutes protons below the pain threshold; it can reduce injection-related discomfort for patients. Further studies are needed to investigate both causes, as well as solutions, to this problem.

A strong point in favor of the use of BoNT-B is its efficacy extended to patients resistant to BoNT-A. While the potential for secondary resistance to BoNT-B remains unclear, it may occur less than with BoNT-A as commercially available BoNT-B manufacturing procedures do not include freeze-drying and the product does not require reconstitution before use. However, switching to a BoNT-B formulation is undesirable from an immunological perspective, given that BoNT-B has a higher immunogenicity than BoNT-A [30,31]. 

Future research should increase studies comparing how BoNT-A and BoNT-B affect the autonomic and motor nervous systems and how long their action on both systems lasts.

## 4. Material and Methods

A PubMed search from 2000 to November 2022 was performed to identify any reports on the use of BoNT-B in focal hyperhidrosis. We detected these articles using the terms “botulinum toxin type B and focal hyperhidrosis”, “botulinum toxin type B and palmar hyperhidrosis”, “botulinum toxin type B and axillary hyperhidrosis”, “botulinum toxin type B and cranio-facial hyperhidrosis”, “botulinum toxin type B and plantar hyperhidrosis”, “botulinum toxin type B and compensatory hyperhidrosis”, and “botulinum toxin type B and residual hyperhidrosis”. Only studies in English were reviewed. All studies that met the criteria were included and summarized in this review. The PRISMA study flowchart is shown in Figure 1. Our search identified 80 records after removing duplicates. After scanning the titles and abstracts, 44 citations were excluded (reasons for exclusion were the following: (1) articles including, but not specifically related to focal hyperhidrosis s = 20; (2) articles including, but not specifically related to BoNT-B = 20; (3) articles not related to humans = 4, with 36 papers eligible for full text analysis. After examining the full text, 14 case–control, case series studies, randomized controlled trials were considered eligible, and then included in this study and 22 excluded, being review or expert opinion manuscripts. 

## Figures and Tables

**Figure 1 toxins-15-00147-f001:**
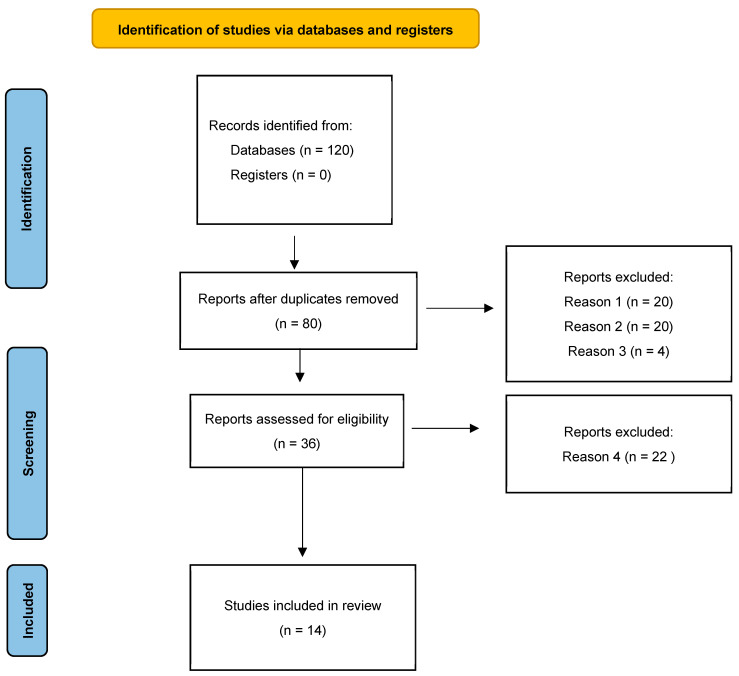
PRISMA flowchart on BoNT-B in focal hyperhidrosis. Research dates range from 1999 to 2022 [32]. Reasons for excluding manuscripts: (1) articles including, but not specifically related to focal hyperhidrosis; (2) articles including, but not specifically related to BoNT-B; (3) articles not related to humans; (4) review or expert opinion manuscripts.

**Table 1 toxins-15-00147-t001:** BoNT-B in axillary hyperhidrosis data from the literature.

First Author Year [Ref.]	Type of Publication	Patients [n]	BoNT-B Doses	Follow Up	Results
Dressler D et al., 2022 [6]	Prospective, single blind controlled clinical study	19	BoNT-B 2000–4000 MU compared to BoNT-A 100 MU contralaterally	2–3 weeks	Similar results between BoNT-B 4000 MU and BonT-A side. No difference in the duration of hyperhidrosis improvement between the axillae treated with BoNT-B 4000 MU and 2000 MU.
Hecht MJ et al., 2004 [7]	Prospective, clinical study	4	250 MU BoNT-B diluted in 2.5 mL of saline was injected subcutaneously into each axilla at 10–15 injection sites	3–12 weeks	Three weeks after the BoNT-B-injection, three of four patients had axillary anhidrosis, and in the fourth patient, axillary sweating was dramatically reduced. After 3 months, axillary sweating recurred in all patients.
Nelson L et al., 2004 [8]		13	5000 MU	4, 8 and 12 weeks	Mean percentage reduction in hyperhidrotic area was 84, 87, and 81% at 4, 8. and 12 weeks, respectively. Patient satisfaction was 100%.
Baumann L et al., 2005 [9]	double-blinded, randomized, pilot study	20	2500 U per axilla	Patients followed until sweating returned to baseline levels.	Duration of action ranged from 2.2 to 8.1 months (mean 5.0 months).
Frasson E et al., 2011 [10]	Paired, single-blinded, randomized study	10	2500 U per axilla vs 50 MU BoNT-A	1 and 2 weeks and 1, 3, and 6 months after treatment	Reduction in axillary sweat production in all treated patients. At 1 and 2 weeks and 1, 3, and 6 months after treatment, sweat weight and area decreased significantly more in the BoNT-B side than in the BoNT-A side.

**Table 2 toxins-15-00147-t002:** BoNT-B in palmar hyperhidrosis data from the literature.

First Author Year [Ref.]	Type of Publication	Patients [n]	BoNT-B Doses	Follow Up	Results
Baumann L. et al., 2005 [11]	Double-blind, randomized, placebo-controlled study	20	5000 U BoNT-B per palm	Participants were followed until sweating returned to baseline levels	Significant difference was found in treatment response at day 30. The duration of action ranged from 2.3 to 4.9 months, with a mean duration of 3.8 months.
Rosell K et al., 2013 [12]	Prospective, clinical study	26	mixed treatment (BoNT-A-Xeomin^®^ plus BoNT-B toxin-Neurobloc^®^) 213 ± 19 U Xeomin^®^ and 264 ± 60 U Neurobloc^®^ over the thenar eminences to avoid muscle weakness	3 weeks	95% of patients were satisfied, evaporation decreased > 50% and Dermatology Life Quality Index score significantly improved from 10.3 to 1.2
Basciani L et al., 2014 [13]	Prospective, clinical study	32	5000 MU BoNT-B	4, 12, and 24 weeks	Significant reduction of palmar sweating was detected after BoNT-B injection after 4, 12, and 24 weeks

**Table 3 toxins-15-00147-t003:** BoNT-B in craniofacial hyperhidrosis data from the literature.

First Author Year [Ref.]	Type of Study	Patients [n]	BoNT-B Doses	Follow Up	Results
Karlqvist M. et al., 2014 [15]	Prospective open study	38	BoNT-B (250 U/mL) with 5 U per injection administered intradermally every 15 mm in a checked pattern covering the entire hyperhidrotic area	2 weeks, then 2 years	87% of the patients were satisfied with the treatment result. In a 2-year follow-up, 74% returned for further treatments after a median time of 5 months.
Cabreus P et al., 2019 [16]	Prospective, clinical study	8	mixed treatment (BoNT-A-Xeomin^®^ plus BoNT-B toxin-Neurobloc^®^) 213 ± 19 U Xeomin^®^ and 264 ± 60 U Neurobloc^®^ over the thenar eminences to avoid muscle weakness	4 weeks	DLQI score was improved for all patients after BoNT-B treatment with a median decrease of 9 points (90% median improvement).
Cantarella G et al., 2010 [17]	Prospective, clinical study	7	BoNT-B mean total dose, 2354 U	12 months	One month after treatment, six of the seven patients reported that gustatory sweating and flushing had resolved, and, in the remaining patient, these symptoms had decreased. At 12 months after treatment, all patients still reported some improvement.

## Data Availability

Not applicable.

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
