# Peer review of "Efficacy and Safety of Botulinum Toxin B in Focal Hyperhidrosis: A Narrative Review"

_toxins, 2023, doi:10.3390/toxins15020147_

Round 1

Reviewer 1 Report

The authors review literature on using botulinum toxin B for treating hyperhidrosis. The selection of papers are based on clearly described criteria. Overall the review is interesting and informative.  It is unfortunate that comparison with type A toxin is not provided except in one of the studies. The manuscript suffers from poor grammar and unclear sentence structure at several places and can be improved tremendously with some more careful editing. 

Author Response

ANSWER: the manuscript has been extensively reviewed to improve the editing

Reviewer 2 Report

Thank you for the opportunity to review the submission. 

1. Although the introduction is rich in information with a current and appropriate citations about BTX B, it is somewhat abrupt with syntax errors, neither presented with a clear focused harm or therapy question, rather presented with a vague aim and no arguments presented on the knowledge gap to conduct this research . Besides, I don't believe a narrative review is a good research design to disseminate knowledge is more acquire way for adapting to one's clinical practice . A systematic review and meta analysis would have been more appropriate. 

2. The search strategy is flawed. Although you have used specific key terms such as BTX B and hyperhidrosis , still your study selection process ( flow chart) shows 40 studies removed because of the reason 1 and 2 i.e not on  hyperhidrosis or BTX B. I would like to see a detailed search strategy as an appendix. 

3. The result section is very rudimentary . Although you described them under axillary, palmar , planar , crania facial, compensatory, residual hyperhidrosis , this doesnt match with your title and efficacy and safety can only be ascertained by conducting a proper systematic review and meta analysis . The result is a mare description of the included studies. 

4. The authors carefully written the discussion section with appropriate comparison with existing studies. However, there is little to no elaboration in regard to the cause, mechanism, effect on various indications and immunogenicity. Moreover, the current section, authors should have delved into the meaning, importance and relevance of their results. It should have been focused on explaining and evaluating what they found, showing how it relates to literature review and research questions, and making an argument in support of their overall conclusion. 

Author Response

  1. Although the introduction is rich in information with a current and appropriate citations about BTX B, it is somewhat abrupt with syntax errors, neither presented with a clear focused harm or therapy question, rather presented with a vague aim and no arguments presented on the knowledge gap to conduct this research . Besides, I don't believe a narrative review is a good research design to disseminate knowledge is more acquire way for adapting to one's clinical practice . A systematic review and meta analysis would have been more appropriate. ANSWER: The introduction has been changed according to the reviewer’s suggestions, fthe objective pursued in this article has been formulated specifically, and the necessity to overcome a gap between knowledge on and therapeutic potentiality of BTX-B has been reported clearly, based on efficacy and safety profile of BTX-B. As regards the choice of carrying out a narrative review, I emphasize that narrative reviews aim to provide an overview of a given topic, generally addressing every aspect of it. They generally investigate on the entire clinical and epidemiological context of a certain pathology/treatment, and aim to provide a basic knowledge of the topic. In this review the main questions to be addressed were: which is the efficacy profile of BTX-B? Which is the best dosage fo every body site?  and these are the typical questions one tries to answer in a narrative review. Otherwise, systematic reviews concentrate the analysis on specific aspects of a given pathology or health intervention, trying to answer a few well-defined clinical questions.  Here is a typical question answered by a RS: “Which of these two therapeutic interventions is more effective in reducing mortality in this type of patient? However, our aim, rather than to compare BTX-B to BTX-A or to any other treatment for hyperhidrosis, was aimed to improve the general knowledge on a such unconventional treatment to hyperhidrosis like BTX-B.

For more details on the appropriate use of systematic or narrative reviews please visit this site: (http://www.cochrane.it/sites/cochrane.it/files/uploads/guidausorevisioni.pdf)

  1. The search strategy is flawed. Although you have used specific key terms such as BTX B and hyperhidrosis , still your study selection process ( flow chart) shows 40 studies removed because of the reason 1 and 2 i.e not on  hyperhidrosis or BTX B. I would like to see a detailed search strategy as an appendix. ANSWER: Usually, that during the selection process of scientific articles, ineligible papers emerge with a frequency comparable to that found by us. However, as suggested by reviewer, a detailed selection of retrieved documents has been reported in appendix, as requested to make the selection process evident.
  1. The result section is very rudimentary. Although you described them under axillary, palmar , planar , crania facial, compensatory, residual hyperhidrosis , this doesnt match with your title and efficacy and safety can only be ascertained by conducting a proper systematic review and meta analysis . The result is a mare description of the included studies.ANSWER: The title of the review has been changed to fit more with descriptive obtained results. The aim of this narrative review is to provide useful Evidence Based Data to improve the knowledge on BTX-B. Actually, it would not have been possible to conduct a systematic review on BTX-B with the aim of making a comparison on the profile and efficacy with BTX-A, due to the scarcity of studies present in the literature.
  2. The authors carefully written the discussion section with appropriate comparison with existing studies. However, there is little to no elaboration in regard to the cause, mechanism, effect on various indications and immunogenicity. Moreover, the current section, authors should have delved into the meaning, importance and relevance of their results. It should have been focused on explaining and evaluating what they found, showing how it relates to literature review and research questions, and making an argument in support of their overall conclusion. 

ANSWER: In discussion section mechanism of action, effect on various indications and immunogenicity of BTX-B have been reported. Moreover considerations on meaning, importance and relevance of  results have been described in dis

Reviewer 3 Report

It is my great pleasure to have an opportunity to review this interesting review paper. In this article, the authors reviewed efficacy and safety of BoNT-B for treatment of focal hyperhidrosis in detail. This article is well-written, interesting, and useful contribution, which I think is entirely suitable for publication in this journal.

Author Response

THANK YOU FOR YOU POSITIVE COMMENTS

Reviewer 4 Report

Please take care in typing - lines 279-280

Please unify the use of 0 before decimals, as you either use the p<.005 variant or the p>0.05 one (lines 360-370). A single variant I believe will be preferable.

Lines 443-444: "the increasing incidence a severity of related side effects"... probably should be "the increasing incidence in the severity of related side effects"

line 457 - with- out should be without

Overall, for a review probably only 27 references are not enough, but since you explained how you got this number I believe it might be considered satisfactory.

Author Response

REV 4

Please take care in typing - lines 279-280  DONE

Please unify the use of 0 before decimals, as you either use the p<.005 variant or the p>0.05 one (lines 360-370). A single variant I believe will be preferable.  DONE

Lines 443-444: "the increasing incidence a severity of related side effects"... probably should be "the increasing incidence in the severity of related side effects" DONE

line 457 - with- out should be without DONE

Overall, for a review probably only 27 references are not enough, but since you explained how you got this number I believe it might be considered satisfactory. OK